# Tumor Mutational Burden for Predicting Prognosis and Therapy Outcome of Hepatocellular Carcinoma

**DOI:** 10.3390/ijms24043441

**Published:** 2023-02-08

**Authors:** Daniela Gabbia, Sara De Martin

**Affiliations:** Department of Pharmaceutical and Pharmacological Sciences, University of Padova, 35122 Padova, Italy

**Keywords:** hepatocellular carcinoma, tumor mutational burden, tumor immune microenvironment, microsatellite instability, neoantigens, liquid biopsy, immune checkpoint inhibitors, immunotherapy

## Abstract

Hepatocellular carcinoma (HCC), the primary hepatic malignancy, represents the second-highest cause of cancer-related death worldwide. Many efforts have been devoted to finding novel biomarkers for predicting both patients’ survival and the outcome of pharmacological treatments, with a particular focus on immunotherapy. In this regard, recent studies have focused on unravelling the role of tumor mutational burden (TMB), i.e., the total number of mutations per coding area of a tumor genome, to ascertain whether it can be considered a reliable biomarker to be used either for the stratification of HCC patients in subgroups with different responsiveness to immunotherapy, or for the prediction of disease progression, particularly in relation to the different HCC etiologies. In this review, we summarize the recent advances on the study of TMB and TMB-related biomarkers in the HCC landscape, focusing on their feasibility as guides for therapy decisions and/or predictors of clinical outcome.

## 1. Introduction

Hepatocellular carcinoma (HCC) accounts for 90% of primary liver malignancies, with about 900,000 new diagnosed cases in 2020 (4.7% of total new diagnosis) [1]. HCC is the second-highest leading cause of cancer death worldwide, causing 8.3% of total cancer-related deaths. A great deal of evidence is suggestive of a possible increase in the number of new diagnoses in the next 20 years [2]. HCC can either develop spontaneously or is more frequently the outcome of chronic liver diseases of different etiologies, such as viral infections (HBV and HCV), alcoholic and non-alcoholic steatohepatitis (ASH and NASH), and cirrhosis. Since transplantation, resection, and ablation are approaches limited to patients with early stage HCC, and most patients are diagnosed at advanced stages or/and have underlying chronic liver disease, finding new curative options to prolong survival and reduce tumor recurrence is a priority in the field of this hepatic cancer. Sorafenib has been used for years for the treatment of advanced-stage HCC according to the Barcelona clinic liver cancer (BCLC) staging system [3]. However, issues related to its efficacy and toxicity have been raised [4], due, among other reasons, to the induction of metabolizing enzymes [5,6,7]. Although the introduction of immunotherapy for advanced HCC has changed the treatment paradigm, not all HCC patients are responsive to these new treatments. Therefore, several strategies are currently being explored to increase the efficacy of HCC pharmacological treatment by adding chemosensibilizers [8] and/or tailored combination therapies [9]. Moreover, in patients treated with immunotherapy, an increase of immune-related adverse events has also been observed [10,11]. In this scenario, the personalization of HCC therapy is becoming a huge priority to maximize clinical benefit, together with the identification of new biomarkers able to select potential responders by efficiently predicting the clinical response to pharmacological treatment.

In this context, many studies have analyzed the relationship between tumor mutational burden (TMB) and immune checkpoint inhibitor (ICI) efficacy in different tumors [12]. The prognostic value of TMB is extremely variable in different cancers due to the different ranges used to define low and high TMB values and the methods for TMB determination in tumoral specimens. HCC is ranked as the 12th among 30 different cancers for the median number of tumor mutations [13]. Lower TMB values are associated with a better prognosis in clear cell renal carcinoma [14], colorectal cancer [15], and prostate cancer [16]. At variance, lower TMB is predictive of worse survival in breast cancer [17], melanoma [18], bladder urothelial carcinoma [19], and non-small cell lung cancer [18].

Many efforts have been focused on unravelling the role of TMB, particularly in relation to the different HCC etiologies, as a reliable biomarker to stratify HCC patients in subgroups likely to be responsive to immunotherapy. In this review, we summarize recent advances in the study of TMB in the HCC landscape.

## 2. TMB Analysis in Tumoral Specimen

TMB, defined as the total number of somatic non-synonymous mutations per coding area of a tumor genome, represents one of the emerging biomarkers of response to ICIs, i.e., anti-Cytotoxic T-Lymphocyte Antigen 4 (CTLA-4), anti-programmed death-1 (PD-1), or PD-ligand 1 (PD-L1) drugs. Physiological mismatch repair (MMR) proteins are devoted to recognizing and correcting errors in mismatched nucleotides. Somatic mutations, miRNA-mediated downregulation, or genetic hypermethylation of MMR proteins can lead to deficiency in MMR, leading to the accumulation of mutations in DNA microsatellites. This condition is defined as high microsatellite instability (MSI). In theory, a higher number of mutations leads to an increase of self-neoantigens and immunogenic recognition, facilitating T-cell response triggered towards tumoral cells (Figure 1). Thus, a higher TMB, an index of great dishomogeneity inside tumoral tissue, may suggest an increased responsiveness to immunotherapy [20,21]. The first evidence of the potential use of TMB as a biomarker to predict ICI efficacy has been described for melanoma, whereas some other studies failed to demonstrate a correlation between TMB and ICI responsiveness [12]. The predictive role of TMB in the context of HCC immunotherapy has not been completely unraveled.

Critical factors in this comprehension are the standardization of (1) TMB cutoffs to stratify patients into low, intermediate, and high classes; (2) methods for TMB calculation to ensure reproducibility between different cancers and laboratories. Some studies have indicated different values to discriminate between high- and low-TMB groups, e.g., using cutoffs to define high TMB either > 3 [22], >10 [23,24], or >20 mutations/megabase [25]. The first method set up to characterize TMB was whole-exome sequencing (WES), which considers non-synonymous mutations in the exomes, excluding germline alterations by subtracting matched normal samples. Since this technique is technically complex and expensive, it has been supplanted by the advent of next-generation sequencing (NGS) [26,27,28]. Recently, liquid biopsy, i.e., the measure of circulating tumor DNA (ctDNA), has also been used for the estimation of TMB. In particular, liquid biopsy has recently gained more attention in HCC prognostic and clinical research due to many advantages, e.g., the possibility of non-invasive and repeated sampling on the same patient, since multiple blood samples can be performed more easily than standard tumor tissue biopsies [29].

An interesting study analyzed tumoral and peritumoral HCC samples and matched peripheral blood mononuclear cells (PBMCs), observing that clonal HCC evolution is driven by the accumulation of somatic mutations and ctDNA may be effective in tracking tumor clonal evolution [30]. Specific microenvironments, e.g., inflammatory background, toxic mutagens and treatments, that could induce a characteristic mutational signature, provide a “selective pressure” driving HCC clonal evolution, and influence the accumulation of somatic mutations (Figure 1). A similar study investigated the correlation between tissue TMB (tTMB) and blood TMB (bTMB), measured as ctDNA, in 136 patients with unresectable HCC enrolled by 4 centers, whose ctDNA profile was analyzed between October 2020 and July 2021 [31]. A high frequency of TP53, CTNNB1, and TERT mutations was observed in their ctDNA profiles. Furthermore, a significant difference was observed between bTMB, whose mean and median were 10.6 and 8.6 mutations/megabase, and tTMB, whose mean and median were 4.8 and 3.0 mutations/megabase, respectively. This discrepancy is likely due to both technical factors, e.g., differences in sampling time, size and location of sequenced genome regions, or algorithms used for TMB calculation, and intrinsic biological mechanisms. In fact, ctDNA could derive from multiple tumor foci, each characterized by a peculiar mutational profile and evolution [31]. This study pointed out that the evaluation of bTMB and ctDNA, besides being reliable indicators of TMB in tumor tissue, may represent more representative markers of HCC evolution, since ctDNA can be derived from multiple tumor foci. The prognostic role of TMB in HCC can be affected by the fact that different HCC clones might coexist in the liver, rendering a single tumor biopsy not representative of a high MSI. A study by Mukai suggested that the collection of multiple samples from tumors should be considered to clarify the real proportion of MSI-H, and that epigenetic aberrations may lead to MSI-H in HCC patients [21]. Another fundamental technical issue in the routine evaluation of TMB in HCC patients is the optimization of sampling, since TMB can be evaluated in both archival and fresh tumoral samples. In this regard, a study by Wong et al. compared both type of samples in order to understand the effect of the sampling technique in TMB evaluation, showing that the use of fresh unfixed tissues is the best choice to obtain reliable results [32].

A prospective, multicenter, observational cohort study, which is currently recruiting patients (NCT04484636), aims to develop a platform to identify frequent targetable mutations and evaluate TMB. The final aim is evaluating the possible use of this platform, called PLATON (Platform for Analyzing Targetable Mutations), for the personalization of therapy in different gastrointestinal cancers (HCC, cholangiocarcinoma, gallbladder carcinoma, pancreatic, and esophagogastric cancers). Both tumor specimens and whole blood samples will be analyzed by NGS to evaluate whether and how patients have been treated according to their genomic profiles. The results of this study will likely be available from mid-2023, hopefully assisting with the therapy personalization of gastrointestinal cancers.

Many efforts have also been focused on the predictive potential of neoantigens, in particular of those able to activate cytotoxic T-cell immune response, since, as already stated, a higher TMB is related to an increased neoantigen formation. An analysis of a cohort of HCC patients of The Cancer Genome Atlas (TCGA) reported that both quantity and quality of TMB and neoantigens were not associated with prolonged survival in patients not receiving immunotherapy, whereas a strong correlation between TMB, the number of predicted neoantigens, and patient survival has been observed in patients treated with ICI [33]. The results of a recent study suggested that only high-affinity neoantigens (HANs) are correlated with an improvement of survival by virtue of their ability to modulate tumor antigen-specific CD8+ cell response. Although this study failed to find a correlation between TMB and neoantigens with overall survival [34], it made clear that the high-HAN group of patients receiving anti-PD1 therapy had a better response than the other patients, probably due to an increased activation of tumor-reactive CD39+/CD8+ T cells compared to low-HAN patients.

To date, many conflicting results regarding the predictive role of TMB as a reliable biomarker of tumor prognosis have been reported. Many studies have focused on the combination of TMB with other specific genetic signatures to predict HCC outcome (Table 1). Although most evidence suggests that TMB may represent a reliable marker of immunotherapy outcome, its predictive value for HCC prognosis is more debated.

A study performed on a cohort of 327 HCC patients tested the predictive value of a novel immune-related risk signature, based on a panel of 5 prognostic immune-related genes, on patients’ prognosis and therapy outcome. No difference in TMB was observed between patients with “high” or “low” risk of HCC progression according to this immune-related signature, and no difference in overall survival was observed between high- and low-TMB groups [49]. At variance, differences in the expression of the five immune-related genes seemed to be predictive of survival and immune response in HCC patients.

## 3. Tumor Immune Microenvironment

Liver immunology is an emerging field of research, since the immune system plays a fundamental role in hepatic physiology and pathology [50,51,52]. In liver cancer, the hepatic tumor microenvironment (TME) consists of a wide variety of cell types, including immune cells, such as tumor-associated macrophages (TAMs), myeloid-derived suppressor cells (MDSCs), tumor-associated neutrophils (TANs), and regulatory T cells (Tregs), but also cancer-associated fibroblasts (CAFs), hepatic stellate cells (HSCs), and endothelial cells (Figure 2). Additionally, proteins present in the extracellular matrix play a role in this context, since they are able to interact with immune cells and modulate their function [53,54,55,56,57,58]. The complex crosstalk occurring among these characters aims to sustain an environment with tolerogenic features towards HCC cells that fosters their immune escape and cancer progression [59]. The immune checkpoint molecules PD-L1, PD-L2, and CTLA-4 play a fundamental role in the regulation of immune response towards cancer cells, by suppressing the activation of protective immune cells, e.g., cytotoxic T cells, and promoting immune surveillance, e.g., by recruiting Tregs [60,61,62].

TAMs, monocyte-derived macrophages (MoMϕs) recruited from the bloodstream in response to chemokine (C-C motif) ligand 2 (CCL2) and macrophage colony-stimulating factor (M-CSF) release, differentiate towards M2-type macrophages with pro-tumoral activity. In consequence of persistent hepatic injury, inflammation causes attempts at tissue repair by activating HSCs and recruiting M2-macrophages, leading to the formation of dysplastic nodules that may undergo neoplastic transition and immune cell reprogramming to prompt cancer immune escape [63,64,65]. The accumulation of macrophages in HCC tissue was also associated with an increase of Tregs via an IL-10 dependent pathway, indicating that TAMs could also sustain tumor progression by affecting intratumoral Tregs, further promoting cancer progression by improving immune escape [66].

MDSCs represent a heterogeneous population of immature myeloid cells, displaying two main phenotypes, i.e., polymorphonuclear MDSC (PMN-MDSC), similar to neutrophils, and monocytic MDSC (M-MDSC), similar to monocytes [67]. Both these phenotypes can induce local immunosuppression and tumor progression, since they are able to hamper CD4+ T cells, CD8+ T cells, natural killer (NK) cells, and the release of cytokines and chemokines involved in angiogenesis and recruitment of immune cells [68]. MDSCs are also able to induce fibrosis and CAF activation, and are involved in sorafenib resistance via FGF1 upregulation [67].

CAFs are important tumor stromal cells, responsible for promoting cancer cell aggressiveness and stemness. By acting on different pathways, they drive cell metabolism toward glycolysis and glutamine reductive carboxylation. These processes are pivotal in sustaining tumor immune escape and increasing angiogenesis [69]. Since CAFs could have different origins, they display different phenotypic traits and functions, which still need to be identified and unraveled [70]. However, the majority of CAFs present in HCC have a pro-tumoral activity and their abundancy is generally correlated with poor prognosis [69]. Some CAF subtypes have been reported to exert different effects on tumor cells. For example, aSMA-expressing MRC5 fibroblasts are reported to induce apoptosis in the LM3 hepatoma cells, although other evidence demonstrates that they favor non-classical epithelial–mesenchymal transition, thus enhancing cell motility and invasiveness [71]. Moreover, the release of miR-199a from mesenchymal stem cell-derived CAFs exerts a chemosensitizing effect of HCC cells towards doxorubicin treatment [72].

Even though neutrophils have been generally recognized to be devoted to host defense towards microorganisms, TANs have been recently identified as orchestrators of the release of cytokines and chemokines with a paracrine protumor or antitumor effect in response to tumoral milieu [73]. CCL2^+^ or CCL17^+^ TANs have been correlated with increased tumor growth, invasion, encapsulation, and differentiation [74], and lower number of these cells in tumors are predictive of longer survival. Moreover, CCL2^+^ or CCL17^+^ TANs could recruit CCR2^+^ macrophages and Tregs through the CCL2 Receptor CCR2 and CCL17–C-C Chemokine Receptor 4 (CCR4).

The intratumoral increase of FoxP3^+^ Tregs has been generally recognized as one of the most effective events in tumor immune escape and associated with poor HCC prognosis. Physiologically, cytotoxic CD8+ T cells recognize neo-antigens produced by mutated cells and target them to destruction, preventing cancer development. The activation of cytotoxic T cells is due to costimulatory molecules that are crucial in reprogramming naïve T-cell metabolism towards an activated state [75]. To escape immune surveillance, tumor cells increase the expression of immune checkpoints that physiologically attenuate immune response against healthy tissues, and also increase the recruitment of Treg cells [66]. A recent study identified two clusters of HCC patients, based on a signature of six costimulatory molecule genes (CMGs), that could help in identifying poor-prognosis patients and their ICI outcome [35]. They observed that the low-risk cluster of patients had a lower TMB, low frequency rate of TP53 mutation, higher immunophenoscore (IPS), IPS-CTLA4, IPS-PD1/PD-L1/PD-L2, and IPS-PD1/PD-L1/PD-L2^+^CTLA4 with respect to the high-risk cluster. This CMG-related signature suggests this cluster of HCC patients could benefit from ICI therapy more than the high-risk one. An analysis of 32 primary HCC tissues collected from patients admitted to the Hospital of Guangdong between May 2019 and November 2020 revealed that PD-L1-positive patients had lower TMB, greater vascular invasion and more advanced BCLC stages than PD-L1-negative patients [44]. Moreover, among the 30 most mutated genes, this study identified TP53, CTNNB1, KMT2D, AXIN1, ALK, and NOTCH1.

Many efforts have been devoted to characterizing HCC according to specific immune subsets, even though the multiple etiologies also complicate the puzzle, introducing other variables to be considered for the development of a predictive model. In general, an increase of CD4+ and CD8+ T cells and M1 macrophages, together with a decrease of Tregs, is associated with a better prognosis, consistent with an improved immune recognition of cancer cells [38,76,77]. A study by Liu et al. investigated the relationship between TMB signature and tumor-infiltrating immune cells. A total of 363 HCC samples were classified into high- or low-TMB groups, and their differences regarding 22 immune cell subtypes were compared [40]. They observed an increased infiltration of resting dendritic cells, eosinophils, and Treg cells in the high-TMB group, as well as a decrease in neutrophil infiltration, leading to a lower overall survival. On the contrary, a higher level of plasmacytoid dendritic cells in the low-TMB group was related to improved overall survival. Thus, high TMB and high frequency of immune-related gene mutation seemed to be predictive of worse HCC prognosis and relapse.

## 4. TMB and Immune TME

Although TMB has been successfully investigated as a prognostic marker for many high-TMB-rate cancers, its usefulness in HCC is challenging due to the lower TMB rate typical of this tumor [13]. Many studies have investigated the relationship between TMB and the modulation of tumoral immunity. A WES analysis of tumoral and peritumoral tissues collected from 100 Chinese HCC patients and 175 HCC patients from the TCGA revealed that HBV-related HCC had a low immune cell infiltration and TP53 mutation, but TMB was not associated with immune infiltration in HBV- and alcoholic-related HCC patients of the TCGA cohort [78]. At variance, the level of copy number variation seemed to be more predictive of immune infiltration than TMB, even if a great heterogeneity among all the analyzed groups of HCC patients was observed.

A study by Xie et al. proposed a novel panel to predict HCC prognosis on the basis of TMB value and immune infiltration [41]. This study analyzed 374 HCC patients, observing a higher infiltration of T-helper (Th) 2, Th17, and gamma-delta T (Tgd) cells in high-TMB patients, and a higher infiltration of Tregs, mucosal-associated invariant T cells (MAIT), and dendritic cells in low-TMB patients, related to better survival in these patients. Another study analyzing TMB and immune infiltration in samples derived from 436 American patients tried to underline specific traits predictive for immunotherapy success by means of the multi-omics approach [42]. This study observed that 10 genes are frequently mutated in HCC, i.e., TP53, TTN, CTNNB1, MUC16, ALB, PCLO, MUC, APOB, RYR2, and ABCA, and 4 of them (CECR7, GABRA3, IL7R, and TRIM16L) are correlated to TMB and prompt antitumor immune infiltration. Moreover, the high-TMB group displayed a great abundance of CD8+ T cells, CD4+ memory-activated T cells, and memory B cells.

Mo et al. observed that CTNNB1 mutations are positively correlated with TMB and prognostic of a better prognosis in HCC patients [43]. Furthermore, CTNNB1-mutated HCC have a high infiltration of NK cells and neutrophils and a downregulation of CD96, suggesting its role in modulating NK-cell recruitment. Moreover, since the stemness of tumoral tissue may influence immune TME, a robust and promising prognostic TMB-clinical-risk nomogram was developed to quantitatively assess stemness characteristics to predict patients’ clinical outcome [22]. Another study observed that patients with high TMB have a worse prognosis, although their tumoral environment is enriched in follicular helper cells and activated NKs, at variance with low-TMB tumors that have an increase of resting DCs [47]. In this study, the authors set up a prognostic nomogram to predict HCC prognosis based on three hubs of differentially expressed immune-related genes (DEIRGs).

Few studies have focused on the investigation of immune-related genes (IRGs) in the prognosis of HCC. A 2021 study based on 365 HCC patients of the TCGA cohort set up an immune-related risk signature consisting of eight IRGs to establish a risk model predicting HCC outcome [45]. This study observed that aberrant expression of IRGs is correlated with cancer development, and the high-risk group showed higher TMB, immune cell infiltration, PD-1, PDL1, and CTLA4 expression, and an intensive immune-related phenotype compared to the low-risk group. Another study assessing DEIRGs in tumoral and normal specimens from the TCGA HCC cohort observed that tumoral tissues have an upregulation of BIRC5, CACYBP, NR0B1, RAET1E, SPINK5, and SPP1, and a downregulation of S100A8, in comparison to healthy tissues. The high-risk group assessed by means of the 7-IRG developed model was also related to a higher TMB value, even though in this cohort of patients TMB did not influence overall survival [46].

A large-scale study analyzing all HCC cases from the TCGA-LIHC and ICGC-LIRI-JP datasets demonstrated that risk score assessed from TMB signature was negatively correlated with the activation of immune cells, suggesting that patients with low risk signature are characterized by an immune-activated phenotype, leading to an overall survival longer than those with high risk [79]. Furthermore, high risk was positively correlated with some target genes of immunotherapy, e.g., PD1, PD-L1, PD-L2, CTLA-4, HAVCR2, and IDO1, suggesting the probable increased effect of ICI in this group of patients.

Other evidence on the link between TMB and immune cells have been provided by a study by Gao at al., which also demonstrated the different expression of many kinds of immune cells in HCC and normal tissues. The high presence of M0 and M2 macrophages, and naïve CD4+ T cells, as well as a poor infiltration of CD8+ T cells, were related to poor prognosis [80]. Their study proposed that four differentially expressed genes, i.e., SQSTM1, ME1, BAMBI, and PTTG1, are independent risk factors of a poor prognosis in HCC patients and high TMB is associated with poor survival.

## 5. TMB in the Different Etiologies of HCC and Its Progression

Many efforts have been focused on the possible relationship between TMB or specific mutation signature and HCC etiology, e.g., viral hepatitis, alcoholic liver disease, non-alcoholic fatty liver disease (NAFLD), and non-alcoholic steatohepatitis (NASH). Most of the evidence comes from viral-related HCC, particularly following HBV infection, and NAFLD/NASH-related HCC, the latter representing an increasing cause of HCC in Western countries. In recent years, preventive strategies have been exploited to reduce etiological agents, e.g., vaccination for HBV [81] and nutraceuticals for NAFLD prevention [82,83]. HBV is able to integrate into the genome of infected hepatocytes, thus representing a driver for genetic instability and HCC development [84]. Previous studies have suggested that both viral integration and viral active replication are two mechanisms involved in HCC initiation and maintenance, even when no active virus replication is present in HCC foci and thus HBV infection is silent [85]. Another study observed that epigenetic age acceleration assessed evaluating DNA methylation (DNAm) age, which reflects the chronological age of tissue, is associated with an immunoactive phenotype but lower TMB. In early stages of HCC, lower DNAm age is related to an increase of HBV expression and higher TMB, and these tumors are more prone to proliferate and develop worse malignancies [86]. Another study also found that hypermethylation of the four genes AJAP1, ADARB2, PTPRN2, and SDK1 was present in promoter regions of HCC tissues, whereas these genes were hypomethylated in their body sequence, suggesting an epigenetic regulation of gene expression involved in hepatocarcinogenesis, as previously observed in glioblastoma [87].

Mutation of specific genes have also been described in plasma biopsies from viral-induced HCC patients. For example, CTNNB1 and TP53 mutations have been found in 15% and 38% of patients with HBV-associated HCC, respectively [88]. These data confirmed those of other studies reporting that TP53 mutations are most frequent in HBV than in HCV-related HCC; 30–40% in HBV vs. 20% of patients with HCV-associated HCC [89,90]. Moreover, other mutated gene patterns which have not been described previously were correlated with HCC in this study, e.g., transmembrane protein 141 (TMEM141), A disintegrin and metalloproteinase with thrombospondin type 1 motif, 9 (ADAMTS9), and adhesion G protein-coupled receptor (ADGRV1). The results of this study suggest the possible use of ctDNA and liquid biopsies for monitoring HCC and personalizing treatment [88].

A study on a Chinese HCC cohort observed that patients with microvascular invasion and Edmondson III-IV grade had higher TP53 mutations, whereas those with hepatic capsule invasion carried TERT mutations. In the highly mutated CTNNB1 group, there were more patients with AFP < 25 µmg/L, Edmondson I-II grade, and non-HBV etiology [91]. A recent study in NAFLD-related cirrhosis and HCC observed an enrichment of hTERT mutations in HCC patients, where rare germline hTERT mutations and shorter telomeres in peripheral blood are also frequent [92].

Since NASH-related HCC has shown an increased incidence in recent years, some studies have recently focused on evaluating the molecular features characterizing HCC patients with this particular etiology [93]. A study performed on 43 NASH-related HCC samples and 43 HCC samples of other etiologies confirmed the previous observation that *TERT* (56%), *CTNNB1* (28%), and *TP53* (18%) were the most frequently mutated genes in NASH-related HCC specimens [94]. Moreover, the tumor suppressor activin A receptor type 2A (*ACVR2A*) had a higher mutation rate in NASH-HCC samples compared to that observed in viral/alcohol-related HCCs (10% vs. 3%). This study hypothesized that the NASH microenvironment promotes genotoxicity and specific nucleotide substitution, leading to a specific mutation signature, called MutSig-NASH-HCC, that has been almost exclusively found in NASH-related HCC samples (16% vs. 2% in viral/alcohol-derived HCCs). NASH-related HCC samples also had a lower CTNNB1 mutation prevalence with respect to HCC from other etiologies. The mutational signature observed in NASH-related HCC involves a panel of genes involved in bile and fatty acid metabolism, oxidative stress, inflammation, and mTOR pathway also controlling lipid biosynthesis [95,96,97].

A study of Wong et al. investigated TMB differences in background cirrhosis according to varying stages of HCC development [24]. They observed that HCC and background cirrhosis patients have low difference in TMB when considered altogether, confirming previously reported evidence [33,98,99,100]. However, when considering background cirrhosis and small early HCC, a significant difference in TMB between them and samples of small- and large-progressed HCC could be demonstrated. Thus, early HCC retains a TMB similar to background cirrhosis, whereas TMB changes significantly in HCC progression irrespective of tumor size, suggesting that TMB is not a reliable prognostic marker for differentiating cirrhotic and dysplastic nodules in early HCC samples.

A recent study reported the detrimental role of high TMB, since it was predictive of poor prognosis in patients who underwent radical hepatectomy after HCC recurrence [101]. Based on risk factors determined by a multivariate analysis, the authors of this study set up a nomogram model which integrates TMB, tumor size, and microvascular invasion to assess recurrence-free survival rate. The authors of this study also suggested that the proposed model may help to predict the clinical outcome of postoperative adjuvant therapy with ICI in high-risk patients. Similarly, a 15-gene specific signature associated with TMB has been proposed to predict HCC prognosis [48]. Another study evaluating ctDNA in 41 HCC patients undergoing surgical resection identified 47 gene mutations in the ctDNA analyzed before surgery. This study demonstrated that NRAS, NEF2L2, and MET mutations are related to a shorter time to recurrence and the median variant allele frequency of mutations in preoperative ctDNA is an independent predictor of recurrence-free survival [102].

One study published in 2021 reported the sequencing of the tumor genome of sarcomatoid HCC, with the aim of deepening the clinical characteristics and the molecular profiles of this very rare HCC subtype [103], whose epidemiology, histopathology, radiology, and clinical features were mostly unknown until the publication of two cohort studies in 2019 [104,105]. This study found specific mutation patterns in sarcomatoid samples with respect to other HCC; in particular, a high rate of rearrangement and homozygous deletion of the tumor suppressor gene CDKN2A, and a high rate of mutation of the genes EPHA5 and FANCM, two other well-known tumor suppressors, and MAP3K1. Moreover, the inhibition of cyclin-dependent kinase 4/6 (CDK4/6) with abemaciclib, ribociclib, and palbociclib has been suggested as a potential therapeutic strategy in this subset of patients.

Another rare and poorly understood primary liver cancer is the combined hepatocellular/cholangiocarcinoma (cHCC-CCA) that has been shown to have a median TMB (2.6 mutations/megabase) similar to that of cholangiocarcinoma (CCA, 2.5 mutations/megabase) and HCC (3.5 mutations/megabase), and also some similar patterns of mutations [25,106]. A machine learning analysis based on TMB was demonstrated to be helpful in the classification of cHCC-CCA patients just by considering the genomic similarities to HCC or CCA without additional clinical–pathological information.

## 6. TMB as Biomarker of Therapy Outcome

The spread of new ICIs and treatment combinations for the treatment of advanced HCC has generated an intense interest in the study of biomarkers able to help patient-personalized therapy selection based on these reliable predictors of efficacy and safety [107]. Since high TMB is likely to produce more neoantigens that in turn promote greater infiltration and diversity of antitumoral immune responses, considering TMB as a biomarker could help to make immunotherapy more effective.

In this regard, TMB, the extent of intratumoral CD8+ T cell infiltration, and PD-L1 expression have been proposed as distinct biomarkers of ICI response [108,109], but poor prognostic response or conflicting evidence are reported regarding their real feasibility. Since these factors are functionally interrelated, it seems that a multifactorial biomarker panel incorporating these and other variables is likely to have more reliable predictive value of ICI therapy outcomes than individual markers [107]. With this aim in mind, many studies have focused on evaluating a panel of genetic signatures that could predict the ICI therapy outcome.

A recent study by Hu et al., exploring the association between immune microenvironment, TMB, mutation signature, and driver-gene mutations in Chinese HCC patients, observed that high values of TMB (>10 mutations/megabase) are associated with increased PD-L1 expression, as well as increased numbers of CD3+ T-cells, CD68+ TAMs, and CD66b+ TANs. This pattern of mutations is the most prone to recurrence and poor overall survival after resection [23].

A meta-analysis evaluating the objective response rates to PD-1/PD-L1 inhibitors in HCC patients from the TCGA cohort observed that there was no correlation with the presence of a viral HBV/HCV-induced cirrhosis. This metanalysis also failed to observe correlation between viral etiology, TMB, and tumor immune microenvironment in response to PD-1 inhibitor therapy, concluding that viral setting should not be used as a driver for decision therapy [110]. As regards HCC recurrence following HCV treatment with direct acting antiviral drugs (DAAs), the correlation with TMB has not yet been investigated to the best of our knowledge. Although DAAs seem to improve HCC patients’ overall survival, conflicting results are reported regarding their effect on HCC recurrence [111,112,113,114,115,116].

On the other hand, the KEYNOTE-158 (NCT02628067), a multicenter, non-randomized, open-label trial, led to the approval of pembrolizumab by the FDA in 2020 for the treatment of unresectable or metastatic tumor in adults and children with high TMB (≥10 mutations/megabase) [117]. A combination of the anti-PD-1 antibody SHR-1210 and the VEGFR2 inhibitor apatinib showed improved survival in patients with advanced HCC and high TMB (>7.2 mutations/megabase) compared to patients with lower TMB [118].

The BIOSTORM study analyzed 188 tumoral tissues from patients treated with sorafenib (83) or placebo (105), and set up a predictive 146-gene signature able to identify 30% of patients, characterized by a beneficial effect of sorafenib treatment in terms of recurrence prevention after tumoral resection [119].

The use of a genetic-based predictive model was also investigated as a predictor of response to infusional fluorouracil, leucovorin, and oxaliplatin (HAIC-FO). The exploratory, randomized, interventional phase III FOHAIC-1 trial identified a model based on a 15-gene mutation signature able to predict 83% of patients benefitting from longer progression-free survival and overall survival, suggesting the usefulness of gene profiling for selecting potential beneficiaries for HAIC-FO treatment with respect to sorafenib [120]. Another study observed that in patients with advanced HCC with portal vein tumor thrombus and FGF21 amplification, TACE with drug-eluting beads (DEB-TACE) plus lenvatinib had a better objective response rate, overall survival, and time to progression than those treated with DEB-TACE plus sorafenib [121], thus confirming the usefulness of WES analysis in TACE decision process.

A case report about a 63-year-old man with intermediate-stage HCC, initially receiving TACE and enrolled to receive the combination of atezolizumab plus bevacizumab after disease progression, showed that PD-L1 expression in tumor-infiltrating immune cells and CD8+ T-cell abundance in the tumor area decreased in the post-progression tumor samples, whereas TMB and MHC class I protein expression increased, suggesting that loss of antigen presentation and neoantigen depletion were not responsible for the resistance to therapy observed in this patient, which could more likely be due to several tumor-intrinsic signatures linked to tumor dedifferentiation [122]. The IMbrave 151 study investigated the use of the combination atezolizumab-bevacizumab in HCC with noteworthy results [123]. The anti-angiogenic drug bevacizumab is known to reduce tumor vascularization, improve perfusion, and reverse VEGF-mediated immunosuppression, inducing the immunosuppressive TME to an immunostimulatory switch. An alternative proposed mechanism involved bevacizumab-induced hypoxia, which increases TMB by stress-induced mutagenesis, and thus creates a hypermutated tumor profile, rendering it more responsive to immunotherapy with atezolizumab [124].

Another study reported the development of a gene-mutation-associated nomogram to predict overall survival in the early HCC state and improve a tailored therapy decision [125]. This study evaluated 695 HCC patients from 4 countries, developing a decision model based on independent variables, i.e., T stage, age, country, and the mutation status of the 4 genes TP53, MACF1, EYS, and DOCK2. The analysis of infiltrated immune cells indicated that the high-risk group had more M0 macrophage cells infiltrated into the TME and a significant decrease of CD8+ T cells with respect to the low-risk group, which was also characterized by higher TP53 mutation. Moreover, they observed that the low-risk group was likely to be ICI-sensitive, particularly benefitting from CTLA-4 blockade. TP53, catenin1 (CTNNB1), titin (TTN), mucin 16 (MUC16), and albumin (ALB) are the most common (top five) mutations in HCC. TP53 and LRP1B mutations were also associated with higher TMB and poor prognosis in another study analyzing HCC patients from the TCGA and Chinese clinical dataset [39]. In Chinese patients, another study observed that the most frequently mutated genes were TP53, TERT, and CTNNB1, responsible for the regulation of P53, Wnt, and telomere repair pathway, respectively [91], and higher TMB was associated with higher CTNNB1 mutation. Similarly, another study assessing mutation frequencies in 81 HCC tumor samples by NGS observed that high TMB is associated with mutations in five specific genes: TP53, CTNNB1, AT-rich interactive domain-containing protein 1A (ARID1A), myeloid/lymphoid or mixed-lineage leukemia (MLL), and nuclear receptor co-repressor 1 (NCOR1), and that high TMB and these associated mutations may represent potentially effective biomarkers for the prediction of ICI therapy outcome [99]. Increased CTBB1 and TP53 mutations was also observed in a study analyzing fine-needle biopsy FFPE specimens collected from 46 HCC patients treated with sorafenib. In this study, TMB was correlated with the clinical response to sorafenib therapy and identified the gene expression signature of TGFa, PECAM1, and NRG1 as a predictor for sorafenib response and progression-free survival [126]. A recent meta-analysis on 17 randomized-controlled trials observed that mutations of TERT, CTNNB1, BRD4, or MLL, and co-mutations in TP53/TERT and TP53/BRD4, were associated with worse survival, and a risk score based on these mutations may be more predictive for survival of ICI-treated HCC patients than TMB [127].

A study by Spahn et al. failed to show difference in the median TMB between responders and non-responders and a correlation between TMB and progression-free survival, concluding that TMB is not useful in identifying ICI responders among HCC patients [128]. This study analyzed HCC patients treated with nivolumab or pembrolizumab in a range of time from August 2015 to December 2019, enrolled by the University Hospital of Tuebingen and Munich (Germany), Vienna (Austria), and Bern (Switzerland). However, the lack of correlation between TMB and ICI outcome observed in this study could be due to the small number of patients (*n* = 15) undergoing WES characterization.

A clinical case report published in 2020 reported for the first time the achievement of a good response to cabozantinib plus nivolumab therapy in an HCC patient with bone metastasis with RET amplification, copy number of 5, TMB ≥ 10 mutations/megabase and high PD-L1 expression [129]. This case report thus demonstrated the potential of combined therapy with cabozantinib and ICI in patients with advanced HCC, high TMB, and bone metastasis.

The most recent retrospective analysis considering all studies investigating the TMB predictive response to immunotherapy [130] included all the studies on PD-1/PD-L1 inhibitor-based monotherapies and combination therapies up to the end of January 2022, which set up a hazard ratio of progression-free survival and overall survival based on the TMB values. This analysis, although concluding that high TMB represents an encouraging biomarker to predict survival in HCC patients treated with immunotherapy, stated that TMB alone is not sufficiently stable as an independent biomarker, and encouraged the exploration of further combination models.

In the future, the results of ongoing clinical studies investigating on the role of TMB in HCC will be available (Table 2), and might help to provide more insight into its potential use as a predictor of therapy outcome and HCC progression.

## 7. Combination of TMB and Other Specific Genetic Biomarkers

In order to set up more predictive and robust biomarkers for HCC, many studies have investigated novel risk scores integrating the analysis of TMB with other specific gene mutation signatures. A recent study by Liu et al. investigated the potential function of Zinc Finger CCHC-Type Containing 17 (ZCCHC17) in HCC, assessing its expression in 90 paraffin-embedded specimens. This study observed that a higher ZCCHC17 expression reflects a higher TMB and MSI in HCC tissues and is also related to higher TP53 mutation. Moreover, these high values were correlated to high histological grade and pathological state, tumor status, and vascular invasion [36].

A study conducted on 590 HCC samples divided patients into three classes according to the distinct molecular subtypes of N6 adenosine methylation (m6A), the main RNA modification involved in tumor proliferation, replication, invasion, and metastasis [131]. According to this classification, higher m6A score was associated with high TMB and worse prognosis. Another study clustered HCC samples into distinct groups according to DNA methylation and built a CpG-based prognostic signature able to precisely predict prognosis [132]. Another recently published study showed that eight m6A-associated small nuclear RNAs (snRNA) are predictors of survival and ICI outcome, and low risk is associated with a low-TMB value [133]. Moreover, another study found that a model based on long non-coding RNA (lncRNAs) is superior to TMB status in predicting HCC patients’ overall survival, even though patients with high TMB also have high risk scores [134]. Additionally, N7-methylguanosine (m7G) mRNA modification is likely to be correlated with many human diseases. A study analyzing the correlation between m7G-modified lncRNAs and HCC overall survival showed a correlation between a 22 m7G-related lncRNA-based risk score, TMB, and the expression of immune checkpoints [135].

microRNA (miRNAs) and lncRNAs have recently gained attention as possible markers for prediction of HCC and therapy outcome. In the last two years, many studies have been published on this topic [136,137,138,139,140]. One of them set up a prediction model based on eight lncRNAs strongly associated with TMB and tumor immune infiltration to predict HCC patients’ prognosis [141]. This study suggests that in high-TMB HCC, the LINC00638/miR-4732-3p/ULBP1 axis is likely to regulate the immune system tumor evasion via PD-L1. Another study suggested that a complex network consisting of 3 lncRNAs, 12 miRNAs, and 21 mRNAs may help in classifying patients into two subgroups and predicting HCC progression independently from TMB status [142]. Another study investigated the connection among single nucleotide polymorphisms (SNPs), TMB, and the epithelial–mesenchymal transition EMT-related lncRNAs, observing a specific signature to predict HCC prognosis, with high TMB and TP53 mutation associated with worse prognosis [143].

Since lactate is an important driving metabolite in cancer progression and immune escape, a study investigated the impact of the lactate-metabolism-related gene signature (LMRGS) score in combination with TMB on HCC patients’ survival [37]. This study analyzed transcriptomic data of TCGA and International Cancer Genome Consortium (ICGC) cohorts, observing that TMB in the TCGA cohort was higher in the LMRGS high-score group. This signature was associated with a shorter overall survival and median survival compared to the low-TMB group. This group of HCC patients with high TMB and LMRGS scores also displayed a high expression of inhibitory immune checkpoint molecules, suggesting that these patients might benefit more from immunotherapy than the low-score patients. Thus, another study observed a positive correlation between TMB and immune cell infiltration scores, and both these scores were independently correlated with immunotherapy response [38]. This study supports the evidence that high infiltration of CD4^+^ T cells, CD8+ T cells, and M1-type macrophages, and low Treg infiltration, were associated with a better prognosis.

A high TMB was also observed in patients displaying mutation of low-density lipoprotein (LDL) receptor-related protein 1B (LRP1B), who had a poor prognosis [39]. Thus, the presence of LRP1B mutation may help in predicting HCC prognosis in those patients with higher TMB and higher expression of human endogenous retrovirus-H long terminal repeat-associating protein 2 (HHLA2), another immune checkpoint gene.

Other specific signatures that have been correlated to TMB, worse overall survival, and disease-free survival are glycolysis-related genes (GRGs) signatures [144] or piroptosis-related gene mutations [145]. Additionally, aberrant expressions of RNA terminal phosphate cyclase like 1 (RCL1) [146] and variation of mRNA expression-based stemness (mRNAsi) index [147] have been correlated with TMB and modification in overall survival. Conversely, another study observed a negative correlation between TMB and ferroptosis-related genes [148].

Additionally, ELMO1, a protein involved in the regulation of SOX10 expression, which is able to induce epithelial–mesenchymal transition through PI3K/Akt signaling, was negatively linked to TMB [149]. This negative correlation with ELMO1 suggests a negative relationship between epithelial–mesenchymal transition and TMB in HCC.

A recently published study identified the NCBP2 gene among the highly mutated genes in high-TMB HCC patients. This gene is involved in the regulation of proliferation, metastasis, and apoptosis, and has been proposed as a novel biomarker of carcinogenesis and cancer progression [150].

## 8. Conclusions

In recent years, TMB has gained more interest as a biomarker to predict cancer progression and therapy outcome. Its predictive value has been observed in patients affected by various solid tumors. In HCC, TMB is quite low with respect to other solid tumors, and its predictivity value is still debated and far from being fully demonstrated. A consensus seems to exist regarding the fact that high TMB is related to reduced survival, although it is likely to be also predictive of an improved response to immunotherapy. A high TMB usually leads to an enhanced production of neoantigens, which could be effectively recognized after restoring the immune response by ICI therapy. However, the results are not conclusive, probably because of the heterogeneity in the definition of the range of high TMB, as well as the etiology of HCC. Moreover, although the evaluation of TMB by measuring ctDNA in liquid biopsies has gained increasing attention in recent years, due to easy sampling and the possibility of following HCC evolution over time, more evidence needs to be collected on their correlation with TMB in HCC tissues. In conclusion, TMB seems to be potentially useful as a form of support for decisions on patients’ therapy, even though this parameter probably needs to be integrated with the assessment of other specific mutation signatures. In this way, a more complex but more reliable score to predict HCC and therapy outcome could be set up. Furthermore, a standardization of TMB assessment in HCC patients is mandatory to render it effective in clinical practice. A harmonization of the use of this marker can be pursued by reducing the variability among the different laboratories in the methods used for TMB calculation, and a proper and rigorous definition of the TMB cutoffs used to stratify patients.

## Figures and Tables

**Figure 1 ijms-24-03441-f001:**
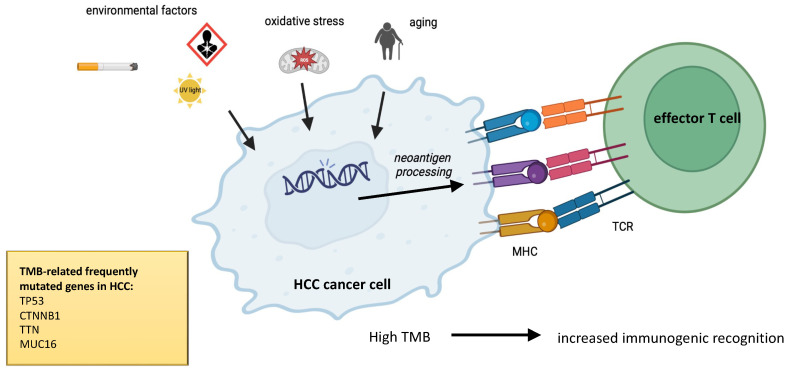
Multiple environmental factors, e.g., UV sunlight, smoke, mutagens, oxidative stress, persistent inflammation, and aging, prompt mutational burden of tumoral cells, increasing neoantigen processing and recognition of neoantigens by effector T cells.

**Figure 2 ijms-24-03441-f002:**
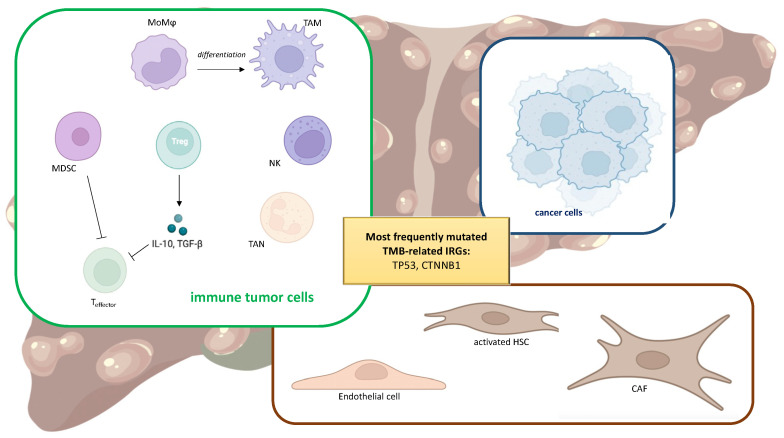
Main cells populating the tumor microenvironment, i.e., monocyte-derived macrophages (MoMϕs) differentiating into tumor-associated macrophages (TAMs), myeloid-derived suppressor cells (MDSCs), tumor-associated neutrophils (TANs), natural killer cells (NKs), regulatory T cells (Tregs), and cytotoxic CD8+ T (Teffector), as well as cancer-associated fibroblasts (CAFs), hepatic stellate cells (HSCs), and endothelial cells. TP53 and CTBBN1 are the two genes most frequently mutated showing a correlation with TMB.

**Table 1 ijms-24-03441-t001:** Most representative studies investigating the correlation between TMB and HCC outcome. The most frequently mutated genes related to high TMB are reported (>14%).

Study	Mutation Pathways	Most Frequently Mutated Genes Associated with High TMB	Population and Database
Hu et al. [23]	High TMB (>10) is associated with increased PDL1 expression, CD3+ T-cell infiltration, and high numbers of CD68+ TAMs and CD66b+ TANs. High-TMB patients display recurrence and poor overall survival after curative resection.	TP53, TSC1, CTNNB1	182 Chinese HCC patients (91.2% HBV-related etiology)
Hu et al. [35]	Low-risk cluster of patients, assessed by six costimulatory molecule gene (CMGs) signature, had a lower TMB, low frequency rate of TP53 mutation, higher immunophenoscore (IPS), IPS-CTLA4, IPS-PD1/PD-L1/PD-L2, and IPS-PD1/PD-L1/PD-L2+CTLA compared to the high-risk cluster.	TP53	50 normal samples and 374 HCC samples The Cancer Genome Atlas (TCGA)
Liu et al. [36]	The high expression of ZCCHC17 is related to AFP, histologic grade, tumor status, vascular invasion, and pathological stage. A high expression of ZCCHC17 was associated with high TMB and microsatellite instability.	TP53	374 HCC patients TCGA LIHC (hepatocellular carcinoma) project
Li et al. [37]	Probability of genetic mutations, overall survival and median survival in the high-LMRGS group were significantly shorter than in the low-LMRGS group. In the high-LMRGS group, the immune microenvironment presented more inhibitory immune cell infiltration (follicular helper T cells and regulatory T cells).	TP53, TTN, CTNNB1	TCGA-HCC dataset used as the training cohort, ICGC-LIRI-JP dataset as validation set
Yang et al. [38]	High immune cell infiltration score was characterized by enhanced activation of immune-related signaling pathways and a significantly higher TMB. Immune cell infiltration score could predict patient responses to immunotherapy independently from TMB.	n.a.	571 HCC patients The Cancer Genome Atlas (TCGA) and International Cancer Genome Consortium (ICGC) cohorts
Liu et al. [34]	CD39+PD-1intCD8+ TILs displayed an effector phenotype and stronger antitumor activity in high-high-affinity neoantigens (HAN) versus low-HAN group.	TP53, CTNNB1, ARID1A	56 patients with HCC in The First Affiliated Hospital of Sun Yatsen University
Mauriello et al. [33]	In cancer patients undergoing immunotherapy, a stronger correlation between TMB, number of predicted neoantigens, and survival was observed.	n.a.	115 Hepatocellular carcinoma (HCC) patients available from The Cancer Genome Atlas (TCGA)
Liu et al. [39]	Low-density lipoprotein (LDL) receptor-related protein 1B (LRP1B) mutation was associated with a higher TMB and higher expression of HHLA2. The prognosis of HCC patients with LRP1B mutation was poor.	LRP1B, TP53, TTN, MUC16, AHNAK2, OBSCN, FLG, PCLO, HMCN1, USH2A, CSMD3, XIRP2, RYR2	361 HCC patients from TCGA 399 cases from International Cancer Genome Consortium (ICGC)
Liu et al. [40]	Higher infiltrating abundance in the high-TMB group correlated with worse OS and hazard risk for high-TMB patients in HCC. CD8+ T cells and B cells were associated with improved survival outcomes. High TMB indicated good HCC prognosis and promoted tumor immune infiltration.	TP53, TTN, CTNNB1, MUC16	376 HCC patients from The Cancer Genome Atlas (TCGA) cohort
Xie et al. [41]	The prognosis of the high-TMB group was worse than that of the low-TMB group (cutoff TMB limit = 4.9).	TP53, TTN, MUC16, CTNNB1, PCLO	374 LIHC patients were downloaded from the TCGA database through the GDC data portal 203 HCC patients from Japan were also downloaded from ICGC
Yin et al. [42]	CECR7, GABRA3, IL7R, and TRIM16L mutations were associated with TMB and immune infiltration, and promoted antitumor immunity in HCC.	TP53, TTN, CTNNB1, MUC16	374 HCC samples and 50 matched normal samples from GDC portal
Mo et al. [43]	CTNNB1 was one of the frequently mutated genes in HCC and highly associated with survival and TMB. CTNNB1 mutation was significantly associated with a better prognosis.	TP53, TTN, CTNNB1, MUC16	260 patients from LIRI-JP, 369 from LICA-FR, and 394 from LINC-JP in ICGC database
Xu et al. [44]	PD-L1 positive patients had more vascular invasion and advanced CCLC stage. PD-L1 positive patients exhibited a lower TMB compared to the PD-L1 negative group. The most frequent driver gene mutations included *TP53, CTNNB1, KMT2D, AXIN1, ALK,* and *NOTCH1*.	TP53, CTNNB1, KMT2D, AXIN1	32 patients with primary HCC who were admitted to Hospital of Guangdong Medical University
Liu et al. [45]	Identification of a specific gene expression signature useful to predict prognosis and stratify patients with different sensitivities to immunotherapy. TMB was higher in the high-risk group than in the low-risk group.	n.a.	597 HCC patients from The Cancer Genome Atlas (TCGA) and International Cancer Genome Consortium (ICGC)
Peng et al. [46]	Identification of an immune signature included seven differentially expressed IRGs (BIRC5, CACYBP, NR0B1, RAET1E, S100A8, SPINK5, and SPP1) to predict HCC patients’ survival and immunotherapy response. The high-risk group had significantly higher TMB than the low-risk group. The high-risk group had higher TMB, and immunotherapy might be more effective in the high-risk group.	TP53, CTNNB1, TTN, MUC16	372 TCGA-HCC samples were used 242 data sets downloaded from GEO (https://www.ncbi.nlm.nih.gov/geo/ accessed on 20 January 2023) database and 232 patients’ data from LIRI-JP of International Cancer Genome Consortium (ICGC) database
Xie et al. [47]	Higher TMB was associated with worse prognosis in HCC patients. Less CD8+ T-cell enrichment was found in patients with higher TMB. The poor prognosis was in accordance with higher TMB and more activated NK cells.	TP53, CTNNB1, TTN, MUC16	LIHC cohort were collected from The Cancer Genome Atlas (TCGA) database GSE14520 dataset; LIRI cohort were acquired from the International Cancer Genome Consortium (ICGC) database
Huo et al. [48]	HCC patients with high TMB had a poor prognosis, and displayed higher proportion of CD8+ T lymphocyte infiltration compared to the low-TMB group.	TP53, TTN, CTNNB1, MUC16, PCLO	801 HCC patients from The Cancer Genome Atlas Liver Hepatocellular Carcinoma (TCGA-LIHC)1 and International Cancer Genome Consortium (ICGC), LIRI-JP)

n.a. not analyzed.

**Table 2 ijms-24-03441-t002:** Ongoing clinical trial investigating the use of TMB as a biomarker for prediction of HCC progression and therapy outcome (downloaded from https://www.clinicaltrials.gov/ accessed on 2 February 2023).

NCT Number	Status	Outcome Measures	Study Population	Type of Study	Number of Enrolled Patients
NCT03236935	Active, not recruiting	Maximum tolerated dose (MTD); dose-limiting toxicities (DLTs) and other adverse events; recommended interventional phase 2 dose (RP2D) of L-NMMA in combination with pembrolizumab; antitumor activity; plasma concentrations of L-NMMA when combined with pembrolizumab	18 years and older	Interventional phase 1b study	12
NCT04042805	Recruiting	TMB performed by NSG in association with ORR and survival after treatment with Sintilimab (PD-1 antibody) combined with Lenvatinib(TKI)	18 to 100 years	Interventional single-arm, single-center, unrandomized, open-label phase II study	36
NCT04484636	Recruiting	Distribution of mutations in HCC Evaluation of relative frequency of targetable mutations (incl. TMB and MSI status) per disease group Number of patients receiving therapies in accordance with their genomic profiles	18 years and older	Prospective, multicenter, observational cohort study with biobanking	200
NCT05240404	Recruiting	Evaluation of TMB in patients undergoing adjuvant toripalimab therapy after curative-intent ablation for HCC recurrence	18 to 75 years	Interventional phase 2	116
NCT04523493	Recruiting	Evaluation of the correlation between TMB and therapy efficacy in advanced HCC patients undergoing toripalimab combined with lenvatinib vs. lenvatinib	18 to 75 years	Prospective, randomized, placebo-controlled, double-blind, multicenter phase III registration clinical study	519
NCT04605731	Recruiting	Evaluation of TMB, response, and survival outcomes in patients treated with durvalumab and tremelimumab after radioembolization	18 years and older	Interventional phase 1b	32
NCT04723004	Active, not recruiting	Correlation between TMB and the efficacy of toripalimab combined with bevacizumab in advanced HCC	18 to 75 years	Prospective, randomized, open-label, parallel-group, active controlled, multi-center phase III	326

## Data Availability

Not applicable.

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
