# Peer review of "Tumor Mutational Burden for Predicting Prognosis and Therapy Outcome of Hepatocellular Carcinoma"

_ijms, 2023, doi:10.3390/ijms24043441_

Round 1
Reviewer 1 Report
The authors in this review tackle a very interesting and a novel topic regarding the role of tumor mutational burden in hepatocellular carcinoma and it role as a predictor for the prognosis and response to immunotherapy. Some minor comments are to be addressed.
1. In the introduction section, line 39 and 40: the authors mentioned" Sorafenib has represented the gold standard therapy for years", this is inaccurate. Sorafenib is not a gold standard in HCC and its use is limited to advanced HCC according to BCLC staging, so please delete this sentence.
2. Under the title TMB analysis in tumor specimen, line 90-92: add details to explain the liquid biopsy method and how it is performed in a non-invasive manner.
3. In the section of TMB analysis in tumor specimen line 157, the term low risk and high risk is not clear. This refers to the risk of developing HCC or risk of failure of therapy or risk of advanced tumor?, please define.
4. The section Tumor immune microenvironment is complex to the reader, it is better to be illustrated by adding a figure.
5. The review is missing data regarding the relation between microRNA (miRNA) and lncRNA (long non-coding RNA) with the TMB in HCC.
6. According to this review, can the authors link the occurrence of HCC following treatment of HCV by direct acting antiviral drugs (DAAs) which is frequently observed with TBM?
7. Some abbreviations are mentioned in the text without explanation of the original word such as PBMCs (line 98), BCLC (line 235), NAFLD/NASH(line 319).
Author Response
The authors in this review tackle a very interesting and a novel topic regarding the role of tumor mutational burden in hepatocellular carcinoma and it role as a predictor for the prognosis and response to immunotherapy. Some minor comments are to be addressed.
1. In the introduction section, line 39 and 40: the authors mentioned" Sorafenib has represented the gold standard therapy for years", this is inaccurate. Sorafenib is not a gold standard in HCC and its use is limited to advanced HCC according to BCLC staging, so please delete this sentence.
We thank the reviewer for this comment and modified the text accordingly.
2. Under the title TMB analysis in tumor specimen, line 90-92: add details to explain the liquid biopsy method and how it is performed in a non-invasive manner.
We thank the Reviewer and modified the text by adding some details on liquid biopsies, as suggested.
3. In the section of TMB analysis in tumor specimen line 157, the term low risk and high risk is not clear. This refers to the risk of developing HCC or risk of failure of therapy or risk of advanced tumor?, please define.
We specified in the text the term “low” and “high” risk as suggested “No difference in TMB has been observed according to this immune-related signature between patients with “high” or “low” risk of HCC progression”
4. The section Tumor immune microenvironment is complex to the reader, it is better to be illustrated by adding a figure.
We are grateful to the Reviewer for the suggestion. We added a figure (Figure 2) in the section 3.
5. The review is missing data regarding the relation between microRNA (miRNA) and lncRNA (long non-coding RNA) with the TMB in HCC.
We thank for this noteworthy suggestion. We added some studies investigating the relation between TMB, miRNAs and lncRNAs (lines 562-582).
6. According to this review, can the authors link the occurrence of HCC following treatment of HCV by direct acting antiviral drugs (DAAs) which is frequently observed with TBM?
We added a paragraph on this issue (lines 451-455).
7. Some abbreviations are mentioned in the text without explanation of the original word such as PBMCs (line 98), BCLC (line 235), NAFLD/NASH(line 319).
We added the explanation of these acronyms in the text.
Reviewer 2 Report
This review article is written well and emphasized that TMB related biomarkers in the HCC landscape, focusing on their feasabilty as guide for therapy decision and/or predictors of clinical outcome and they recommended that a standardization of TMB measure in HCC patients is mandatory to standardize the use of this marker and render it effective in the clinical practice. The table explored the all the possibilities of correlation between TMB and HCC outcome. Minor comment: The figure should have more clarity.
Author Response
on their feasabilty as guide for therapy decision and/or predictors of clinical outcome and they recommended that a standardization of TMB measure in HCC patients is mandatory to standardize the use of this marker and render it effective in the clinical practice. The table explored the all the possibilities of correlation between TMB and HCC outcome. Minor comment: The figure should have more clarity.
We modified the Figure 1 to make it more clearly understandable, also adding some additional information regarding the most frequently mutated genes related to TMB.
Reviewer 3 Report
In the manuscript titled "Tumor mutational burden for predicting prognosis and therapy outcome of hepatocellular carcinoma" Daniela Gabbia and Sara De Martin conducted a comprehensive review on TMB and its different roles in TME, differentiation of HCC etiologies, and as a biomarker. The manuscript is well written, structured, and comprehensive. It was a pleasure to read it. I have only a view comments/suggestions that should be addressed by the authors.
- The value of the manuscript would be increased if figures/tables were provided on the most frequently mutated genes of TMB (figure/table) or on immune cell interaction in TME in association with TMB (figure).
- The value of the manuscript could be increased if a table/summary of ongoing studies investigating the use of TMB as a biomarker were added.
- “Furthermore, a standardization of TMB measure in HCC patients is mandatory to standardize the use of this marker and render it effective in the clinical practice.” (l. 592) What standardization of TMB measure would the authors suggest? Please comment.
Author Response
In the manuscript titled "Tumor mutational burden for predicting prognosis and therapy outcome of hepatocellular carcinoma" Daniela Gabbia and Sara De Martin conducted a comprehensive review on TMB and its different roles in TME, differentiation of HCC etiologies, and as a biomarker. The manuscript is well written, structured, and comprehensive. It was a pleasure to read it. I have only a view comments/suggestions that should be addressed by the authors.
- The value of the manuscript would be increased if figures/tables were provided on the most frequently mutated genes of TMB (figure/table) or on immune cell interaction in TME in association with TMB (figure).
We thank the Reviewer for the careful revision, and we added some details on most frequently mutated genes in the Table, when available in the cited study, and also in the Figure 1. Moreover, to improve the comprehension of tumor immune microenvironment we added a figure in the section 3.
- The value of the manuscript could be increased if a table/summary of ongoing studies investigating the use of TMB as a biomarker were added.
We thank the Reviewer for this suggestion and added a table to summarize the ongoing clinical studies investigating the role of TMB as biomarker and its correlation with therapy outcome and patients’ survival.
- “Furthermore, a standardization of TMB measure in HCC patients is mandatory to standardize the use of this marker and render it effective in the clinical practice.” (l. 592) What standardization of TMB measure would the authors suggest? Please comment.
We thank the Reviewer and add a short paragraph about this point in the conclusion.